# Volatile Compositions of *Panax ginseng* and *Panax quinquifolium* Grown for Different Cultivation Years

**DOI:** 10.3390/foods12010136

**Published:** 2022-12-27

**Authors:** Yejin Kim, Jung-Woo Lee, Ick-Hyun Jo, Nayeong Kwon, Donghwi Kim, Jong-Wook Chung, Kyong-Hwan Bang, Jeehye Sung

**Affiliations:** 1Department of Food Science and Biotechnology, Andong National University, Andong 36729, Gyeongbuk, Republic of Korea; 2Department of Herbal Crop Research, National Institute of Horticultural and Herbal Science (NIHHS), Rural Development Administration (RDA), Eumseong 27709, Chungbuk, Republic of Korea; 3Department of Industrial Plant Science and Technology, Chungbuk National University, Chungbuk 28644, Cheongju, Republic of Korea

**Keywords:** *Panax ginseng*, *Panax quinquefolium*, volatile composition, HS-SPME/GC-MS, chemometric analysis

## Abstract

The present study examined the volatile profiles of *Panax ginseng* (Korean ginseng) and *Panax quinquefolium* (American ginseng) grown for different cultivation years by using HS-SPME/GC-MS and determined the key discriminant volatile compounds by chemometric analysis including principal component analysis (PCA), hierarchical cluster analysis (HCA), and partial least squares-discrimination analysis (PLS-DA). Fifty-six compounds, including forty terpenes, eight alcohols, one alkane, one ketone, and one furan, were identified in the ginseng roots. The chemometric results identified two major clusters of American ginseng and Korean ginseng cultivars with distinct volatile compositions. The volatile compounds in fresh white ginseng roots were affected by the species, but the influence of different cultivation ages was ambiguous. The major volatile components of ginseng roots are terpenes, including monoterpenes and sesquiterpenes. In particular, panaginsene, ginsinsene, α-isocomene, and caryophyllene were predominant in Korean ginseng cultivars, whereas β-farnesene levels were higher in American ginseng. The difference in volatile patterns between Panax ginseng and Panax quinquefolium could be attributed to the composition of sesquiterpenes such as β-panaginsene, ginsinsene, caryophyllene, and β-farnesene.

## 1. Introduction

Ginseng root, a perennial semi-shade plant belonging to the genus *Panax* of the family Araliaceae, is one of the most important medicinal crops in East Asia [1]. Ginseng is usually cultivated under semi-shade conditions (approximately 30% sunlight) because of its sensitivity to high temperatures. Long term cultivation (3–6 years) is required to grow mature roots under optimal conditions [2]. *Panax ginseng* Meyer (Korean ginseng) and *Panax quinquifolium* L. (American ginseng), which are major commercial ginseng varieties, have been used as valuable resources for more than 2000 years [3]. Many studies have reported numerous pharmaceutical effects of ginseng root, including anti-fatigue [4], immunomodulatory [5], antioxidant, and anti-diabetic activities [6], which could be attributed to bioactive compounds such as ginsenosides, polysaccharides, and phytosterols [3].

Ginseng root is an ingredient in various salads, soups/stews, candy, tea, and other beverages [7]. The unique aroma and flavor of ginseng root are favorable for health-conscious consumers. The compounds responsible for the aroma and flavor characteristics of ginseng root are mainly volatile. The volatile compositions, including sesquiterpenes, monoterpenes, alcohols, and aldehydes, differ depending on the *Panax* species, cultivation age, and biological factors [8]. It has been reported that the characteristics and qualities of various plant food materials could be derived from their volatile compounds [9,10]. However, there is a lack of information on whether the volatile composition of white ginseng root is versatile for *Panax* species, cultivars, and cultivation years.

In previous studies, analytical methods based on solid-phase extraction techniques have been used to isolate volatile compounds due to trace amounts of volatile compounds in ginseng roots, which often results in drawbacks such as the possibility of sample contamination and the low extraction yield of analytes during the extraction step [11,12]. Headspace solid-phase microextraction (HS-SPME) is a rapid, simple, solvent-free technique. It is considered an appropriate tool for sample preparation in volatile compound analysis of numerous food samples [13]. Although a few reports determined volatile components generated from ginseng fruits or red ginseng juice by analyzing HS-SPME coupled with gas chromatography mass spectrometry (GC/MS) [14,15], the volatile profile of fresh white ginseng roots by HS-SPME/GC-MS has not been investigated.

Therefore, the objectives of this study were to determine the volatile components of four different white ginseng roots grown for 3–6 years, using HS-SPME/GC-MS, and to examine the influence of species, cultivars, and cultivation year on the volatile composition of white ginseng roots using chemometric analysis.

## 2. Materials and Methods

### 2.1. Chemicals

Methanol (≥99.9% purity, HPLC grade) was purchased from Honeywell (Charlotte, NC, USA). α-Pinene, 2-pentyl furan, isoamyl butyrate, 1-hexanol, caryophyllene, methyl salicylate, and caryophyllene oxide were purchased from ChemFaces (Wuhan, Hubei, China). 3-Decanone was obtained from Tokyo Chemical Industry (Tokyo, Japan). α-Guaiene and β-sesquiphellandrene were purchased from Toronto Research Chemicals (North York, NY, USA). β-farnesene and €-2-nonen-1-ol were purchased from Santa Cruz Biotechnology (Santa Cruz, CA, USA). β-Pinene, 1-undecene, octanal, 2-heptanol, 2-nonanone, 1-octanol, viridiflorol, 2,3-butanediol, 2-methylbutyl 2-methylbutyrate, and a mixture of n-alkanes (C7-C30) were obtained from Sigma-Aldrich (St. Louis, MO, USA).

### 2.2. White Ginseng Root Samples

*P. ginseng* and *P. quinquifolium*, which belong to the same family, Araliaceae, and are closely related, were used in this study. These *Panax* species were grown in the fields of the National Institute of Horticultural and Herbal Science of Rural Development Administration, Eumsung, Chungbuk Province, Republic of Korea, which is located at 36°94′29″ N, 127°75′21″ E. *P. ginseng* consisted of three commercially cultivated cultivars in Korea including ‘Cheonryang’ [16], ‘Yunpoong’ [17], and ‘Gumpoong’ [18], and *P. quinquifolium* was a local variety collected in Wisconsin, United States. In November 2013, 2014, and 2015, the seeds of *P. ginseng* and *P. quinquifolium* were sown annually in the same field, and four-, five-, and six-year-old ginseng roots were harvested simultaneously in October 2019. Each biological replicate consisted of three randomly selected ginseng roots from each plot. After harvest, the samples were washed with tap water, quenched by adding liquid nitrogen, and then stored at −80 °C until further analysis.

### 2.3. Volatile Compound Analysis

#### 2.3.1. HS-SPME Conditions

Fresh white ginseng roots were manually ground in liquid nitrogen using a mortar and pestle to prepare ginseng pastes. Two grams of ginseng paste sample were placed into a 20 mL headspace vial containing internal standard (2-methylbutyl 2-methylbutyrate, the final concentration of 1 μg/mL in the vial) and 3 mL of distilled water. The stock solutions of the internal standard were prepared by dissolving the pure chemical in methanol to give a concentration of 10 mg/mL. An SPME fiber coated with divinylbenzene/polydimethylsiloxane (DVB/PDMS, 65 μm film thickness, Supelco, Bellefonte, PA, USA) was used in combination with a PAL AOC-6000 Autosampler (Shimadzu, Columbia, MD, USA). The SPME fiber was preconditioned at 250 °C for 10 min to remove contaminants. The sample vial was equilibrated at 50 °C for 40 min in a thermostatic autosampler tray. After equilibration, the fiber was exposed for 20 min at 50 °C with agitation speed of 250 rpm, and then thermally desorbed for 10 min into a splitless GC injection port at 240 °C.

#### 2.3.2. GC-MS Analysis

GC-MS analysis was performed using a Shimadzu QP2020 GC-MS model (Shimadzu Corporation, Kyoto, Japan) equipped with an HP-FFAP capillary column (50 m × 0.32 mm i.d., 0.65 μm film thickness; Agilent Technologies, Wilmington, DE). The mass detector was operated in electron impact mode at an ionization energy of 70 eV with a scanning mass range of 30–500 m/z. The GC oven temperature program was started at 50 °C for 2 min, 50–80 °C at a rate of 5 °C/min, then 80–250 °C at a rate of 3 °C/min, and finally held at 250 °C for 5 min. Helium was used as the carrier gas at a constant flow rate of 1.0 mL/min. The transfer line temperature was maintained at 230 °C. A series of n-alkane (C7-C30) standards were employed to determine the linear retention index (RI) of each compound. The volatile components of the sample were identified by matching the mass spectra from the NIST library (ver.14) and the RIs of authentic standards. When authenticated standards were unavailable, tentative identifications were based on the NIST library (ver.14) and a comparison of RIs from the literature. The relative content of each volatile compound in the samples was obtained by calculating peak area ratio between the analyte and internal standard (2-methylbutyl 2-methylbutyrate) in each sample.

### 2.4. Statistical Analyses

The raw dataset was exported to the SIMCA-P-17.0 software package (Umetrics, Umea, Sweden) for multivariate analysis, including PCA, HCA, and PLS-DA. The PLS-DA model was validated with cross-model validation and permutations (*n* = 100) by describing R^2^Y and Q^2^Y values. R^2^X and R^2^Y characterize the explanatory rate of the model for x (cultivars) and y (volatile compounds) variables, respectively, and Q^2^ reflects the predictive ability of the model. The differences between R^2^Y and Q^2^ should not be larger than 0.3, which means that the established model does not overfit [19]. In addition, the Q^2^ and R^2^ values on the permuted datasets should be lower than the values in the actual dataset [20,21]. The variable importance in the projection (VIP) scores was employed to rank the importance of volatile compounds in the prediction model and to point out those with the highest discrimination ability (VIP score > 1.0) between different ginseng roots. The potential discriminant compounds for the separation between groups were visualized as a heatmap using an online server, MetaboAnalyst 5.0 (http://www.metaboanalyst.ca, accessed on 1 December 2022), based on the following criteria: fold changes (FC) >1.5 or <0.667, and *p* < 0.05, Student’s *t*-test.

## 3. Results

### 3.1. Volatile Compounds of Four Different White Ginseng Roots Grown for 3–6 Years

The relative contents of volatile components in fresh white ginseng roots grown for 3–6 years were determined using HS-SPME/GC-MS, as shown in Table 1 and Figure 1. Fifty-six compounds were identified in the samples, including forty terpenes, eight alcohols, one alkane, one ketone, and one furan. These compounds have been reported to be major and typical volatile compounds in ginseng roots [8,11]. The volatile compounds in ginseng roots varied depending on the *Panax* species, cultivars, and cultivation ages. Although there were different volatile compound profiles between the two *Panax* species, the variations in the volatile profiles according to different cultivation ages in the same type were ambiguous. Among the Korean ginseng cultivars, “Gumpoong” and “Cheonryang” had a higher abundance of alcohols such as methyl alcohol, ethanol, 1-hexanol, and 1-octanol compared to “Yunpoong”. Korean ginseng cultivars (“Yunpoong,”“Gumpoong,” and “Cheonryang”) had relatively higher amounts of volatile compounds than American ginseng. The largest group of ginseng-released volatile compounds was terpenes, mainly composed of monoterpenes (e.g., α-pinene and β-pinene) and sesquiterpenes (e.g., ginsinsene, α-isocomene, β-panasinsene, β-elemene, calarene, β-farnesene, and α-neoclovene). Among the sesquiterpene compounds, ginsinsene, β-panasinsene, calarene, α-neoclovene, β-myrcene, and caryophyllene were higher in Korean ginseng cultivars than in American ginseng. Panaxene, valerena-4,7(11)-diene, cedrene-V6, β-neoclovene, bicyclogermacrene, δ-panasinsine, and ginsenol were only detected in the Korean ginseng groups. In contrast, the concentration of β-farnesene was significantly higher in American ginseng than in others. 3-Carene, α-bisabolene, β-bisabolene, and β-sesquiphellandrene were found only in the American ginseng groups. Meanwhile, the amounts of several aldehydes, such as octanal and 2-nonenal, were higher in American ginseng than in Korean ginseng cultivars. The levels of alkanes, ketones, and furan were not significantly different between the groups. It has been reported that plant-derived numerous sesquiterpenes, alcohols, and derived metabolites are highly valued for their desirable odor and flavor characteristics [22]. Our results showed that the profiles of volatile compounds varied between Korean ginseng cultivars and American ginseng, which may influence the perceived aroma and flavor qualities of each white ginseng root.

### 3.2. Chemometric Analysis

Chemometric approaches, such as HCA, PCA, and PLS-DA, enable metabolite profiling datasets obtained from various food samples to be classified according to their geographical origin and processing methods [23,24,25]. In the present study, unsupervised PCA, HCA, and supervised PLS-DA were conducted to elucidate the influence of species, cultivar, cultivars, and cultivation year based on the volatile composition of white ginseng roots.

Unsupervised PCA and HCA were performed using the volatile profiling data of the ginseng roots to reveal the same sample clustering pattern, provide an overview of the trend, and determine putative outliers. Different color groups reflect each of the Korean ginseng cultivars and American ginseng, and closer distances indicate similar volatile composition. As shown in Appendix A, the PCA score plot showed well-separated clusters among American ginseng, “Yunpoong,” and “Gumpoong”/“Cheonryang,” while the age differences in cultivation were not clear. It is clear that the American and Korean ginseng groups were clustered separately. The first two principal components (PCs) represented 47.3% of the total variance (PC1 = 35.7% and PC2 = 11.6%). In addition, the influence of the Panax species and cultivars on each cultivation year based on volatile compound data was also analyzed by PCA and HCA (Figure 2) because the plot of all ginseng groups was ambiguous. The results indicated that four groups were clearly separated in the PCA score plot of 4- (PC1 = 46.9%, PC2 = 19.0%), 5- (PC1 = 41.6%, PC2 = 14.0%), and 6-year-old roots (PC1 = 42.4%, PC2 = 16.1%), except for 3-year-old roots (PC1 = 44.0, PC2 = 19.7%), with no significant sample outliers. Even though the volatile compositions of 3-year-old roots were well-distinguished between the Korean ginseng and American ginseng groups, “Gumpoong” was not completely separated from “Cheonryang”. HCA was performed to elucidate the similarities and dissimilarities of the volatile profiles of different samples. The HCA dendrogram also indicated that the American ginseng group clustered with the Korean ginseng groups. However, the cultivation ages were not clustered on each ginseng root in the PCA score plot, which means the volatile composition of the ginseng roots were not affect by different cultivation ages (Appendix A). These results reflect the dissimilarity between American ginseng and Korean ginseng cultivars.

Supervised PLS-DA was conducted to calculate models that differentiate between groups and select the specific marker compounds responsible for the different ginseng group-dependent clustering based on VIP values above one. The PLS-DA score plot of all ginseng groups based on volatile compounds showed a distinct separation between Korean ginseng cultivars and American ginseng, although there was no clear clustering among Korean ginseng cultivars (Appendix A). In the PLS-DA score plot of the ginseng samples for each cultivation year, the four different groups could be distinguished well (Appendix A). The PLS-DA loading plot revealed that the American ginseng group in 3-, 4-, 5-, and 6-year-old roots was positioned on the right side of the plot and was characterized by relatively high concentrations of 3-carene, β-farnesene, β-sesquiphellandrene, octanal, and 2-nonenal (Figure 3). Monoterpenes such as β-pinene and β-myrcene appeared to be linked to “Yunpoong” cultivar. Most of the discriminative compositions in “Cheonryang” were characterized by the presence of 1-hexanol, 1-ocatanol and ethanol. Although the clustering between “Gumpoong” and “Cheonryang” in 3- and 4-year-old roots was not clear, “Gumpoong” had relatively high levels of 2-nonen-1-nol and (E)-4-decenal in 5- and 6-year-old roots, respectively. Korean ginseng cultivars with 3-, 4-, 5, and 6-year-old roots were relatively located on the left part of the loading plot, and were characterized by the presence of panaginseng, β-panasinsene, ginsinsene, α-neoclevene, β-neoclevene, α-isocomene, caryophllene, and cedrene-V6. To statistically validate these PLS-DA models, permutation testing was performed (Appendix A). The R^2^ and Q^2^ values of the permutation test with each PLS-DA revealed that the models did not overfit the data, thus contributing to a valid model. The recommended values for good fitting of models have been described as R^2^Y intercept <0.3 and Q^2^Y intercept <0.05 [26,27]. Differential volatile compounds were selected according to the criteria of *p* < 0.05 and VIP score >1 from the PLS-DA model (Appendix A). Heatmap cluster analysis was also performed based on the discriminant volatile compounds between the Panax species and cultivars (Appendix A). Two major clusters of American ginseng and Korean ginseng cultivars were identified with distinct patterns of different volatile abundances.

### 3.3. Major Discriminant Volatile Composition of Four Different White Ginseng Roots

The profiles of key discriminant volatile compounds identified in the four different ginseng groups by chemometric data analysis methods are shown in Figure 4. Korean ginseng cultivars had higher relative concentrations of key discriminant volatile compounds than American ginseng cultivars. The profiles of volatile compounds were similar in Korean ginseng cultivars, including “Yunpoong,”, “Gumpoong” and “Cheonryang,” whereas American ginseng released a different volatile pattern. Panaginsene, β-panasinsene, ginsinsene, α-isocomene, and α-neoclovene were predominant in Korean ginseng cultivars, but only a minor amount was present in American ginseng. Cedrene-V6, β-neoclovene, and caryophyllene were identified only in Korean ginseng cultivars. In addition, the level of β-farnesene was higher in American ginseng than in Korean ginseng cultivars. Our data are consistent with a previous study that demonstrated higher concentrations of (E)-caryophyllene and β-neoclovene and lower levels of β-farnesene in the dried roots of *P. ginseng* Meyer than in *P. quinquifolium* L. [11]. These findings imply that differences in volatile patterns between Korean ginseng cultivars and American ginseng were mainly determined by the composition of sesquiterpene hydrocarbons, such as β-panaginsene, ginsinsene, caryophyllene, and β-farnesene. A previous study reported that panaginsene, β-panaginsene, and ginsinsene are major typical sesquiterpene hydrocarbons isolated from the roots of *P. ginseng* Meyer [28]. In addition, it has been demonstrated that the hydrocarbon fraction of sesquiterpenes enhances the citrus, spicy, woody, and hay/straw aspects of a non-aromatized reference beer flavor [29]. Although the odor and flavor activity values for the major discriminant volatile compounds of ginseng roots were not examined in the current study, sesquiterpenes such as caryophyllene usually have a low odor threshold, meaning that they would release enough odor and flavor at low concentrations [30]. Therefore, the present study suggests that these sesquiterpene hydrocarbon profiles may influence the different aroma and flavor characteristics of Panax species and cultivars.

## 4. Conclusions

In the present study, chemometric analysis of the volatile composition of fresh white ginseng roots analyzed by HS-SPME/GC-MS was used to discriminate between Korean ginseng cultivars and American ginseng. GC-MS data showed that the volatile compounds in the ginseng roots varied depending on the species, cultivar, and cultivation age. Unsupervised PCA and HCA showed that Korean ginseng cultivars and American ginseng could be divided into two clusters, which were determined by the different volatile compositions of each ginseng root. Moreover, the supervised PLS-DA model could be used to identify several sesquiterpenes (e.g., β-panaginsene, ginsinsene, α-isocomene, caryophyllene, and β-farnesene) as the key discriminant volatile compounds of different white ginseng roots. Although further study is still needed on the influence of these compounds on the specific aroma and flavor characteristics of ginseng root, our current study suggests that the sesquiterpene hydrocarbon profile of white ginseng root may play an important role in the aroma and flavor qualities of *Panax* species and cultivars.

## Figures and Tables

**Figure 1 foods-12-00136-f001:**
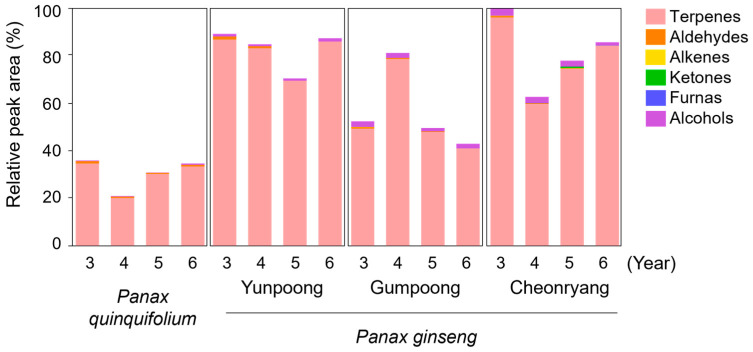
Profiles of volatile compounds present in the four different white ginseng roots grown for 3–6 years. The concentrations of volatile compounds are relatively indicated as percentages based on the total peak area (100) in the 3-year-old ‘Cheonryang’ cultivar. Data are provided as mean values (*n* = 3 biological replicates).

**Figure 2 foods-12-00136-f002:**
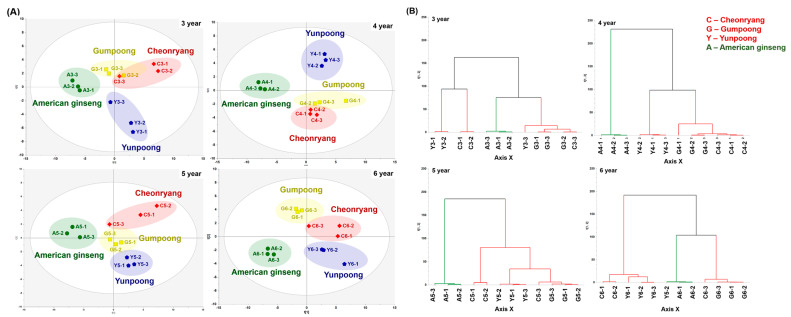
Discriminant analysis of four different white ginseng roots grown for 3–6 years based on volatile compound data obtained from GC-MS. (**A**) PCA score plot and (**B**) HCA dendrogram.

**Figure 3 foods-12-00136-f003:**
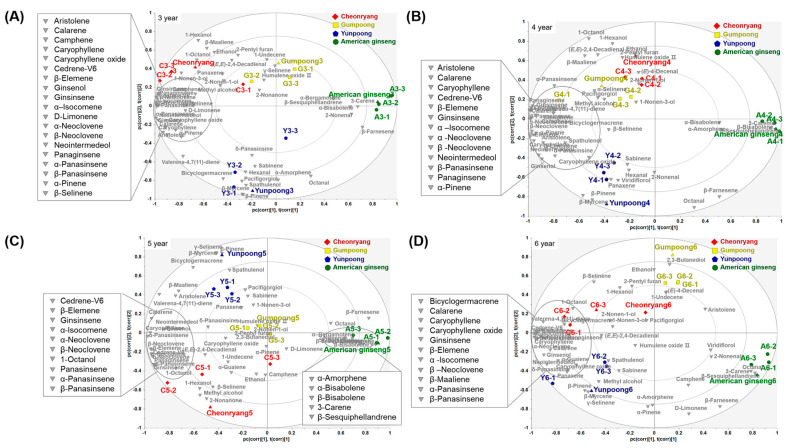
Loading plot of partial least-squares discriminant analysis for four different white ginseng roots based on volatile compound data obtained from GC-MS. (**A**) 3−year−old, (**B**) 4−year−old, (**C**) 5−year−old, and (**D**) 6−year−old ginseng roots.

**Figure 4 foods-12-00136-f004:**
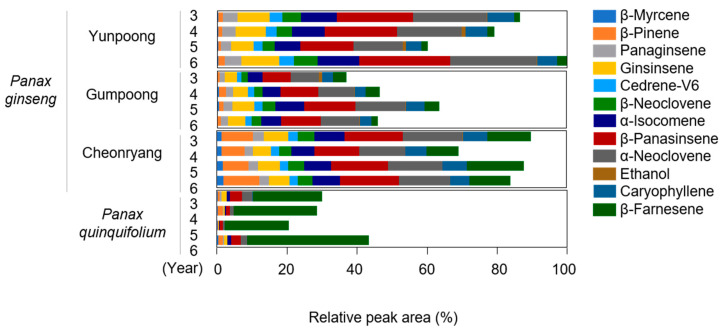
Profiles of the major discriminant volatile compounds identified in the four different white ginseng roots grown for 3–6 years. The tentative concentrations of volatile compounds are relatively indicated as percentages based on the total peak area (100) in the 6-year-old ‘Yunpoong’ cultivar. Data are provided as mean values (*n* = 3 biological replicates).

**Table 1 foods-12-00136-t001:** Composition of volatile compounds in four different white ginseng roots grown for 3–6 years (μg/kg).

Compound	RT ^A^	RI ^B^	American Ginseng				Yunpoong				Gumpoong				Cheonryang				Identification ^C^
			3	4	5	6	3	4	5	6	3	4	5	6	3	4	5	6	
Alcohols																			
Methanol	14.14	918	0.07 ± 0.03	0.11 ± 0.10	0.09 ± 0.03	0.13 ± 0.07	0.20 ± 0.15	0.18 ± 0.08	0.10 ± 0.00	0.21 ± 0.09	0.34 ± 0.04	0.21 ± 0.09	0.12 ± 0.02	0.09 ± 0.00	0.27 ± 0.13	0.20 ± 0.10	0.23 ± 0.09	0.16 ± 0.03	MS, TI
Ethanol	14.95	950	nd ^D^	nd	nd	nd	nd	nd	nd	0.02 ± 0.03	0.06 ± 0.04	0.02 ± 0.03	0.03 ± 0.04	0.21 ± 0.16	0.06 ± 0.05	0.20 ± 0.04	0.22 ± 0.27	0.03 ± 0.03	MS, TI
1-Hexanol	31.93	1372	nd	nd	nd	0.01 ± 0.03	nd	nd	0.02 ± 0.02	0.03 ± 0.04	0.13 ± 0.04	0.06 ± 0.06	0.09 ± 0.03	0.05 ± 0.01	0.20 ± 0.06	0.14 ± 0.02	0.11 ± 0.05	0.05 ± 0.02	MS, TI, FI
1-Octanol	40.85	1577	nd	nd	nd	0.02 ± 0.03	nd	nd	0.12 ± 0.03	0.17 ± 0.09	0.13 ± 0.02	0.21 ± 0.05	0.18 ± 0.10	0.16 ± 0.03	0.34 ± 0.03	0.22 ± 0.03	0.25 ± 0.20	0.14 ± 0.12	MS, TI, FI
2,3-Butanediol	42.10	1610	nd	nd	nd	nd	nd	nd	nd	nd	nd	nd	0.02 ± 0.03	0.05 ± 0.03	nd	nd	nd	nd	MS, TI, FI
1-Nonen-3-ol	44.47	1668	nd	nd	nd	nd	0.01 ± 0.02	nd	0.01 ± 0.02	nd	0.02 ± 0.03	0.01 ± 0.02	0.01 ± 0.02	nd	0.08 ± 0.03	nd	nd	0.02 ± 0.03	MS, TI
2-Nonen-1-ol	47.13	1739	nd	nd	nd	nd	nd	nd	nd	nd	nd	nd	0.02 ± 0.03	nd	0.03 ± 0.06	0.01 ± 0.01	nd	0.02 ± 0.04	MS, TI, FI
Spathulenol	62.23	2190	nd	nd	nd	nd	0.13 ± 0.01	0.11 ± 0.01	0.05 ± 0.04	0.05 ± 0.09	0.03 ± 0.05	0.16 ± 0.07	nd	0.02 ± 0.03	nd	nd	nd	nd	MS, TI
Aldehydes																			
Hexanal	20.29	1111	nd	nd	nd	nd	0.05 ± 0.01	0.01 ± 0.01	nd	nd	nd	nd	nd	nd	0.02 ± 0.03	nd	nd	nd	MS, TI
Octanal	29.60	1322	0.20 ± 0.03	0.15 ± 0.07	0.06 ± 0.08	0.16 ± 0.07	0.28 ± 0.03	0.20 ± 0.03	0.02 ± 0.03	nd	0.01 ± 0.02	nd	nd	nd	0.10 ± 0.17	nd	nd	nd	MS, TI, FI
2-Nonenal	41.04	1584	0.15 ± 0.05	0.03 ± 0.05	0.07 ± 0.08	0.07 ± 0.12	0.07 ± 0.13	0.08 ± 0.14	nd	nd	nd	nd	nd	nd	0.04 ± 0.07	nd	nd	nd	MS, TI
(*E,E*)-2,4-Decadienal	51.96	1872	nd	nd	nd	nd	nd	nd	0.02 ± 0.02	0.02 ± 0.04	0.14 ± 0.05	0.13 ± 0.04	0.06 ± 0.05	nd	0.08 ± 0.14	0.07 ± 0.01	0.04 ± 0.08	0.04 ± 0.06	MS, TI, FI
(*E*)-4-Decenal	57.16	2028	nd	nd	nd	nd	nd	nd	nd	nd	nd	nd	nd	0.02 ± 0.04	nd	0.08 ± 0.07	nd	nd	MS, TI
Alkenes																			
1-Undecene	21.69	1144	nd	nd	nd	nd	nd	nd	0.01 ± 0.02	nd	0.04 ± 0.01	nd	nd	0.01 ± 0.01	0.02 ± 0.03	nd	0.04 ± 0.04	nd	MS, TI, FI
Ketones																			
2-Nonanone	34.09	1421	nd	nd	nd	nd	nd	nd	nd	nd	nd	nd	nd	nd	0.01 ± 0.01	nd	0.18 ± 0.13	0.02 ± 0.03	MS, TI, FI
Furnas																			
2-Pentyl furan	26.38	1251	nd	nd	nd	nd	nd	nd	nd	nd	0.04 ± 0.01	nd	0.01 ± 0.02	0.01 ± 0.01	0.04 ± 0.04	0.04 ± 0.00	nd	0.01 ± 0.01	MS, TI, FI
Terpenes																			
α-Pinene	17.69	1038	0.39 ± 0.17	0.14 ± 0.07	0.43 ± 0.32	0.58 ± 0.14	0.74 ± 0.19	0.75 ± 0.06	0.43 ± 0.07	0.53 ± 0.06	0.40 ± 0.11	0.71 ± 0.12	0.29 ± 0.06	0.22 ± 0.11	0.82 ± 0.26	0.57 ± 0.07	0.45 ± 0.04	0.48 ± 0.15	MS, TI, FI
Camphene	19.46	1087	0.07 ± 0.05	0.02 ± 0.03	0.10 ± 0.08	0.15 ± 0.04	0.09 ± 0.03	0.16 ± 0.02	0.08 ± 0.02	0.10 ± 0.03	0.07 ± 0.04	0.19 ± 0.06	0.07 ± 0.02	0.06 ± 0.03	0.18 ± 0.11	0.11 ± 0.05	0.12 ± 0.01	0.11 ± 0.04	MS, TI
β-Pinene	21.19	1130	0.29 ± 0.26	0.06 ± 0.08	0.29 ± 0.19	0.13 ± 0.06	2.21 ± 0.77	1.55 ± 0.09	1.35 ± 0.07	1.85 ± 0.26	0.19 ± 0.05	0.26 ± 0.04	0.44 ± 0.22	0.12 ± 0.03	0.37 ± 0.12	0.17 ± 0.02	0.25 ± 0.06	0.28 ± 0.06	MS, TI, FI
Sabinene	21.61	1142	nd	nd	nd	nd	0.01 ± 0.01	0.01 ± 0.02	0.01 ± 0.02	0.02 ± 0.02	nd	nd	nd	nd	nd	nd	nd	nd	MS, TI
3-Carene	22.93	1174	0.35 ± 0.25	0.16 ± 0.06	0.12 ± 0.10	0.13 ± 0.03	nd	nd	nd	nd	nd	nd	nd	nd	nd	nd	nd	nd	MS, TI
β-Myrcene	23.14	1176	0.07 ± 0.07	0.01 ± 0.01	0.05 0.01	0.04 ± 0.01	0.36 ± 0.11	0.33 ± 0.02	0.29 ± 0.04	0.28 ± 0.04	0.04 ± 0.01	0.11 ± 0.02	0.10 ± 0.03	0.04 ± 0.00	0.08 ± 0.03	0.05 ± 0.01	0.05 ± 0.02	0.04 ± 0.00	MS, TI
D-Limonene	25.27	1224	0.08 ± 0.05	nd	0.09 ± 0.08	0.11 ± 0.02	0.12 ± 0.04	0.17 ± 0.01	0.05 ± 0.01	0.09 ± 0.01	0.09 ± 0.04	0.14 ± 0.03	0.01 ± 0.02	0.01 ± 0.02	0.18 ± 0.09	0.08 ± 0.07	0.06 ± 0.05	0.03 ± 0.05	MS, TI
Panaginsene	35.25	1445	0.06 ± 0.05	nd	0.04 ± 0.05	0.11 ± 0.02	0.55 ± 0.09	0.56 ± 0.01	0.49 ± 0.07	0.65 ± 0.12	0.43 ± 0.10	0.55 ± 0.14	0.40 ± 0.06	0.30 ± 0.10	1.01 ± 0.21	0.61 ± 0.03	0.78 ± 0.32	0.88 ± 0.28	MS, TI
Panaxene	36.02	1465	nd	nd	nd	nd	nd	0.18 ± 0.15	0.09 ± 0.16	0.26 ± 0.11	0.05 ± 0.08	0.05 ± 0.09	0.03 ± 0.06	nd	0.08 ± 0.14	nd	0.04 ± 0.07	0.15 ± 0.14	MS, TI
Ginsinsene	36.55	1475	0.22 ± 0.09	0.05 ± 0.05	0.09 ± 0.054	0.30 ± 0.03	1.26 ± 0.18	1.28 ± 0.04	1.10 ± 0.14	1.44 ± 0.26	1.02 ± 0.21	1.29 ± 0.29	0.94 ± 0.09	0.69 ± 0.18	2.28 ± 0.52	1.33 ± 0.05	1.78 ± 0.81	1.91 ± 0.53	MS, TI
Cedrene-V6	39.20	1538	nd	nd	nd	nd	0.49 ± 0.10	0.51 ± 0.02	0.46 ± 0.04	0.59 ± 0.11	0.39 ± 0.11	0.51 ± 0.19	0.36 ± 0.06	0.28 ± 0.08	0.87 ± 0.22	0.51 ± 0.02	0.64 ± 0.36	0.74 ± 0.24	MS, TI
α-Isocomene	40.18	1561	0.21 ± 0.12	0.02 ± 0.03	0.08 ± 0.14	0.20 ± 0.03	1.63 ± 0.33	1.58 ± 0.05	1.40 ± 0.13	1.78 ± 0.37	1.20 ± 0.26	1.68 ± 0.61	1.08 ± 0.16	0.89 ± 0.22	2.51 ± 0.63	1.53 ± 0.09	1.94 ± 1.05	2.15 ± 0.60	MS, TI
β-Panasinsene	40.59	1571	0.57 ± 0.24	0.21 ± 0.04	0.22 ± 0.14	0.72 ± 0.09	3.52 ± 0.53	3.39 ± 0.19	2.67 ± 0.28	3.47 ± 0.66	2.37 ± 0.48	3.05 ± 0.76	2.25 ± 0.33	1.71 ± 0.44	5.41 ± 1.00	3.16 ± 0.19	4.31 ± 2.06	4.54 ± 1.22	MS, TI
Viridiflorol	41.04	1584	nd	nd	nd	0.05 ± 0.09	nd	0.20 ± 0.18	nd	nd	nd	nd	nd	nd	nd	nd	nd	nd	MS, TI, FI
β-Maaliene	41.09	1585	nd	nd	nd	0.01 ± 0.03	nd	nd	0.16 ± 0.05	0.20 ± 0.03	0.21 ± 0.07	0.23 ± 0.08	0.09 ± 0.08	0.12 ± 0.02	0.17 ± 0.15	0.13 ± 0.06	0.08 ± 0.04	0.12 ± 0.06	MS, TI
δ-Panasinsine	41.80	1601	nd	nd	nd	nd	0.14 ± 0.03	0.19 ± 0.03	0.09 ± 0.08	0.20 ± 0.07	0.12 ± 0.02	0.15 ± 0.06	0.07 ± 0.06	0.02 ± 0.04	nd	0.05 ± 0.09	0.09 ± 0.15	0.08 ± 0.14	MS, TI
α-Guaiene	42.26	1613	nd	nd	nd	nd	nd	nd	nd	0.13 ± 0.12	nd	nd	nd	nd	nd	nd	0.05 ± 0.09	0.14 ± 0.07	MS, TI, FI
Aristolene	42.31	1614	nd	0.05 ± 0.05	nd	0.08 ± 0.07	0.18 ± 0.06	0.18 ± 0.01	0.15 ± 0.03	0.05 ± 0.09	0.13 ± 0.04	0.16 ± 0.06	0.09 ± 0.02	0.13 ± 0.07	0.16 ± 0.07	0.12 ± 0.01	0.08 ± 0.09	nd	MS, TI
α-Bergamotene	42.33	1615	0.02 ± 0.04	nd	0.04 ± 0.04	nd	nd	nd	nd	nd	nd	nd	nd	nd	nd	nd	nd	nd	MS, TI
β-Elemene	42.82	1626	0.09 ± 0.03	0.04 ± 0.03	0.02 ± 0.03	0.13 ±0.01	1.19 ± 0.28	1.52 ± 0.10	1.10 ± 0.09	1.57 ± 0.41	0.72 ± 0.16	1.35 ± 0.66	0.71 ± 0.18	0.82 ± 0.25	1.07 ± 0.96	1.04 ± 0.16	1.43 ± 0.79	1.68 ± 0.47	MS, TI, FI
Calarene	43.34	1639	0.21 ± 0.22	0.38 ± 0.33	0.14 ± 0.16	0.67 ± 0.11	1.67 ± 0.49	1.40 ± 0.07	1.31 ± 0.21	1.36 ± 0.20	1.33 ± 0.29	1.59 ± 0.29	0.87 ± 0.10	0.94 ± 0.24	1.61 ± 0.59	1.19 ± 0.18	1.15 ± 0.53	1.15 ± 0.40	MS, TI
Valerena-4,7(11)-dien	43.97	1656	nd	nd	nd	nd	2.27 ± 0.107	1.49 ± 0.12	1.81 ± 0.38	1.59 ± 0.45	1.21 ± 0.30	2.73 ± 1.57	1.17 ± 0.19	1.16 ± 0.07	1.43 ± 0.60	0.90 ± 0.20	1.09 ± 0.58	1.16 ± 0.0	MS, TI
β-Farnesene	45.13	1683	7.20 ± 3.26	3.79 ± 1.27	4.97 ± 1.39	4.14 ± 0.62	2.43 ± 0.19	3.43 ± 0.26	1.91 ± 0.48	2.59 ± 1.01	0.39 ± 0.14	0.89 ± 0.30	0.86 ± 0.30	0.79 ± 0.04	0.60 ± 0.17	0.38 ± 0.04	0.42 ± 0.27	0.32 ± 0.09	MS, TI, FI
α-Panasinsene	45.55	1695	0.02 ± 0.04	nd	0.01 ± 0.02	0.12 ± 0.01	1.21 ± 0.33	0.68 ± 0.59	1.03 ± 0.09	1.15 ± 0.29	0.82 ± 0.17	1.32 ± 0.58	0.75 ± 0.13	0.62 ± 0.12	1.51 ± 0.44	0.90 ± 0.08	1.11 ± 0.60	1.26 ± 0.33	MS, TI
α-Neoclovene	45.78	1701	0.41 ± 0.19	0.12 ± 0.04	0.21 ± 0.02	0.61 ± 0.08	3.06 ± 0.48	3.27 ± 0.11	2.73 ± 0.34	3.56 ± 0.75	2.29 ± 0.52	3.00 ± 0.94	2.12 ± 0.35	1.65 ± 0.41	5.15 ± 1.24	2.93 ± 0.29	3.86 ± 1.965	4.39 ± 1.21	MS, TI, FI
Caryophyllene	46.80	1729	nd	nd	nd	0.04 ± 0.04	1.15 ± 0.33	1.43 ± 0.05	1.28 ± 0.11	1.48 ± 0.58	0.67 ± 0.19	1.11 ± 0.51	0.65 ± 0.20	0.64 ± 0.13	1.16 ± 0.51	0.93 ± 0.09	1.31 ± 0.66	1.61 ± 0.50	MS, TI, FI
α-Bisabolene	47.87	1760	0.23 ± 0.21	0.04 ± 0.07	0.24 ± 0.21	nd	nd	nd	nd	nd	nd	nd	nd	nd	nd	nd	nd	nd	MS, TI
β-Bisabolene	47.91	1761	0.50 ± 0.86	1.07 ± 0.08	1.94 ± 0.72	2.09 ± 0.07	nd	nd	nd	nd	nd	nd	nd	nd	nd	nd	nd	nd	MS, TI
β-Neoclovene	47.95	1760	nd	nd	nd	nd	0.86 ± 0.12	0.96 ± 0.04	0.72 ± 0.09	0.99 ± 0.24	0.59 ± 0.15	0.77 ± 0.27	0.54 ± 0.10	0.41 ± 0.10	1.38 ± 0.29	0.75 ± 0.07	0.92 ± 0.52	1.13 ± 0.32	MS, TI
α-Selinene	48.54	1778	nd	nd	nd	nd	nd	nd	nd	nd	nd	0.63 ± 0.32	nd	nd	nd	nd	nd	nd	MS, TI
β-Selinene	48.58	1777	nd	nd	nd	nd	0.36 ± 0.33	0.48 ± 0.42	nd	nd	nd	nd	nd	0.33 ± 0.10	0.82 ± 0.24	0.54 ± 0.05	0.39 ± 0.50	0.85 ± 0.25	MS, TI
γ-Selinene	48.58	1779	nd	nd	nd	0.05 ± 0.04	nd	nd	0.48 ± 0.06	0.73 ± 0.19	0.37 ± 0.08	nd	0.34 ± 0.08	nd	nd	nd	nd	nd	MS, TI
Bicyclogermacrene	48.93	1786	nd	nd	nd	nd	0.89 ± 0.55	0.59 ± 0.11	0.91 ± 0.31	0.55 ± 0.31	0.32 ± 0.12	1.07 ± 0.93	0.27 ± 0.09	0.44 ± 0.02	0.19 ± 0.13	0.21 ± 0.07	0.25 ± 0.07	0.29 ± 0.14	MS, TI
α-Amorphene	49.39	1797	0.02 ± 0.04	0.12 ± 0.22	0.19 ± 0.13	0.05 ± 0.02	0.05 ± 0.04	nd	nd	0.06 ± 0.02	nd	nd	0.01 ± 0.02	0.01 ± 0.03	nd	nd	nd	0.02 ± 0.03	MS, TI
β-Sesquiphellandrene	49.70	1809	0.11 ± 0.19	0.22 ± 0.02	0.48 ± 0.24	0.30 ± 0.01	nd	nd	nd	nd	nd	nd	nd	nd	nd	nd	nd	nd	MS, TI, FI
Pacifigorgiol	56.17	1997	nd	nd	nd	nd	0.05 ± 0.04	nd	0.01 ± 0.02	0.01 ± 0.01	0.01 ± 0.02	0.05 ± 0.06	nd	0.01 ± 0.01	nd	nd	nd	nd	MS, TI
Caryophyllene oxide	58.46	2069	nd	nd	nd	nd	0.17 ± 0.03	0.15 ± 0.01	0.04 ± 0.07	0.08 ± 0.08	0.05 ± 0.09	0.05 ± 0.08	nd	nd	0.25 ± 0.06	0.08 ± 0.07	0.04 ± 0.07	0.05 ± 0.09	MS, TI, FI
Humulene oxide II	60.23	2125	nd	nd	nd	nd	nd	nd	nd	0.12 ± 0.21	0.11 ± 0.19	0.27 ± 0.23	0.29 ± 0.05	0.04 ± 0.07	0.27 ± 0.47	0.27 ± 0.24	nd	nd	MS, TI
Neointermedeol	62.65	2201	nd	nd	nd	nd	0.12 ± 0.02	0.13 ± 0.00	0.09 ± 0.01	0.13 ± 0.07	0.05 ± 0.04	0.10 ± 0.04	0.02 ± 0.04	nd	0.15 ± 0.02	0.06 ± 0.01	0.08 ± 0.07	0.14 ± 0.05	MS, TI
Ginsenol	63.63	2234	nd	nd	nd	nd	0.23 ± 0.05	0.25 ± 0.04	0.11 ± 0.03	0.16 ± 0.08	0.11 ± 0.05	0.18 ± 0.05	0.05 ± 0.04	0.02 ± 0.02	0.31 ± 0.06	0.11 ± 0.02	0.10 ± 0.09	0.19 ± 0.05	MS, TI

^A^ Retention time on the HP-FFAP capillary column; ^B^ Retention indices were determined on HP-FFAP capillary column using n-alkanes (C7−C30) as external reference; ^C^ Identification with FI (fully identified using authentic standard), MS (mass spectrum consistent with that from the NIST library), TI (tentatively identified based on the NIST library and the literature); ^D^ Not detected.

## Data Availability

Data are contained within the article or Appendix A.

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
