# Peer review of "Volatile Compositions of Panax ginseng and Panax quinquifolium Grown for Different Cultivation Years"

_foods, 2022, doi:10.3390/foods12010136_

Round 1

Reviewer 1 Report

Dear Authors,

The manuscript "Volatile compositions of Panax ginseng and Panax quinquifolium grown for different cultivation years" reports the volatile profiles of Panax ginseng (Korean ginseng) and Panax quinquefolium (American ginseng) grown for different cultivation years (3-6) by using HS-SPME/GC-MS and to determine the key discriminant volatile compounds by chemometric analysis including PCA, HCA, and PLS-DA.

The manuscript resulted well written, the methodological approach is innovative, and the content of the manuscript resulted interesting.

However, the English language should be improved throughout the manuscript by a native speaker.

Comments:

-   Line 81. Were the samples grown with the same condition in different years?

-   Table 1. Footnotes are missing. Which is the difference between “nd” and “-“?

-   Figure 2. In the caption “(A) PCA score plot and (B) HCA dendrogram.” is repeated.

-   Figure 2. The resolution is very low, and the text of figure 2b is illegible.

Author Response

Dear Authors,

The manuscript "Volatile compositions of Panax ginseng and Panax quinquifolium grown for different cultivation years" reports the volatile profiles of Panax ginseng (Korean ginseng) and Panax quinquefolium (American ginseng) grown for different cultivation years (3-6) by using HS-SPME/GC-MS and to determine the key discriminant volatile compounds by chemometric analysis including PCA, HCA, and PLS-DA.

The manuscript resulted well written, the methodological approach is innovative, and the content of the manuscript resulted interesting.

However, the English language should be improved throughout the manuscript by a native speaker.

We welcome and thank you for your comments regarding our manuscript. We truly appreciate the efforts and comments put forth by the reviewers and have made the necessary corrections and changes based on those comments in order to improve the scientific integrity of the paper. Enclosed, please find the revised manuscript. English was proofread by a native speaker (via Editage; https://www.editage.co.kr/ and any grammatical or syntax errors were corrected (please see a proofreading certificate).

Comments:

-   Line 81. Were the samples grown with the same condition in different years?

Response) P. ginseng and P. quinquifolium used in our study were grown in the same field at the same area. Therefore, we would like to say that there is no difference in the climate and soil environment in which the materials were grown. Reflecting the reviewer's opinion, we specified the latitude and longitude position of the test field in the manuscript.

-   Table 1. Footnotes are missing. Which is the difference between “nd” and “-“?

Response) “-“ means “not detected (nd)” and we changed “-“ into “nd”.

-   Figure 2. In the caption “(A) PCA score plot and (B) HCA dendrogram.” is repeated.

Response) We deleted the repeated caption.

-   Figure 2. The resolution is very low, and the text of figure 2b is illegible.

Response) We tried to make the high resolution of figure 2b.

Reviewer 2 Report

The manuscript, entitled "Volatile compositions of Panax ginseng and Panax quinquifolium grown for different cultivation years", tried to investigate the volatile profiles of Panax ginseng (Korean ginseng) and Panax quinquefolium (American ginseng) grown for different cultivation years by using HS-SPME/GC-MS and determine the key discriminant volatile compounds by chemometric analysis including PCA, HCA, and PLS-DA. This research is a relatively novel. However, some issues still need to be improved or explained.

1. The abstract could be strengthened by the addition of quantitative data.

2. Line 16: “PCA, HCA, and PLS-DA” These abbreviations should be expressed in their full name.

3. Line 30: In general, references are numbered starting from 1. Please note the format of the references in this manuscript.

4. Line 91: “were” should be “was”.

5. Line 92: Detailed internal standard concentrations should be listed.

6. Line 94: Why did you choose the SPME fiber coated with divinylbenzene/polydimethylsiloxane? Generally, the difference of SPME fiber materials will greatly affect the extraction and adsorption of volatile flavor compounds in samples.

7. Line 144-160; Table 1: Volatile compound content should be expressed in the form of Mean ± standard deviation. It is difficult to distinguish the differences in volatile compounds of samples simply by reacting their contents as the form of average value.

8. Table 1: What do these symbols mean such as “nd” and “-” ?

9. Figure 2: The quality of the pictures should be improved.

10. Why are volatile compounds different in ginseng roots at different cultivation ages? It is suggested to add relevant discussion in the manuscript.

11. Line 279-280: How does age of cultivation affect the volatile compounds of the ginseng roots white ginseng? Specific results should be enumerated.

Author Response

The manuscript, entitled "Volatile compositions of Panax ginseng and Panax quinquifolium grown for different cultivation years", tried to investigate the volatile profiles of Panax ginseng (Korean ginseng) and Panax quinquefolium (American ginseng) grown for different cultivation years by using HS-SPME/GC-MS and determine the key discriminant volatile compounds by chemometric analysis including PCA, HCA, and PLS-DA. This research is a relatively novel. However, some issues still need to be improved or explained.

Thank you for your comments regarding our manuscript. We truly appreciate the efforts and comments put forth by the reviewers and have made the necessary corrections and changes based on those comments in order to improve the scientific integrity of the paper. Enclosed, please find the revised manuscript.

1. The abstract could be strengthened by the addition of quantitative data.

Response) We modified the abstract with the addition of quantitative data.

2. Line 16: “PCA, HCA, and PLS-DA” These abbreviations should be expressed in their full name.

Response) We originally mentioned the full name in the subsection of 2.4 Statistical analysis. As you recommended, we added the full name in Line 16 and removed the repeated full name in the 2.4. subsection.

3. Line 30: In general, references are numbered starting from 1. Please note the format of the references in this manuscript.

Response) As you recommended, we modified the proper format of the references.

4. Line 91:“were”should be “was”.

Response) English was proofread by a native speaker (via Editage; https://www.editage.co.kr/ and any grammatical or syntax errors were corrected (please see a proofreading certificate).

5. Line 92: Detailed internal standard concentrations should be listed.

Response) We used 2-methylbutyl 2-methylbutyrate to give the final concentration of 10 μg/mL in vial). In line 104-106, we added the detailed internal standard concentration.

6. Line 94: Why did you choose the SPME fiber coated with divinylbenzene/polydimethylsiloxane? Generally, the difference of SPME fiber materials will greatly affect the extraction and adsorption of volatile flavor compounds in samples.

Response) The SPME fiber coated with divinylbenzene/polydimethylsiloxane has been generally used for the extensive extraction of ginseng volatile compounds (1, 2). Thus, this study used the same fiber with previous studies.

1) Lee S-J, Moon TW, and Lee JH. 2010. Increase of 2-furanmethanol and maltol in Korea red ginseng during explosive puffing process. Journal of Food Science, 75(2); 147-151.

2) Shen H, Wei T, Zhang Z, Zheng Q, Guo R, Jiang H, Zhang G, and Zheng J. 2020. Discrimination of five brands of instant vermicelli seasonings by HS-SPME/GC-MS and electronic nose. J Food Sci Technol ; 57(11): 4160-4170.

7. Line 144-160; Table 1: Volatile compound content should be expressed in the form of Mean ± standard deviation. It is difficult to distinguish the differences in volatile compounds of samples simply by reacting their contents as the form of average value. 

Response) Thank you for pointing this out. We modified the form of mean ± standard deviation in Table 1.

8. Table 1: What do these symbols mean such as “nd”and “-”?

Response) “-“ means “not detected (nd)” and we changed “-“ into “nd”.

9. Figure 2: The quality of the pictures should be improved.

Response) We tried to make the high resolution of figure 2b.

10. Why are volatile compounds different in ginseng roots at different cultivation ages? It is suggested to add relevant discussion in the manuscript.

Response) Thank you for pointing this out. In this study, the different in volatile compounds of ginsengs were not clear depending on cultivation ages. As you can see in Fig. S1, it is hard to distinguish the groups between the Panax species and cultivars. We divided into each cultivation ages and further analyzed the influence of the Panax species and cultivars by PCA and HCA to find out the clear discrimination among the sample groups. We mentioned about this in line 209-216. Please understand this point.

11. Line 279-280: How does age of cultivation affect the volatile compounds of the ginseng roots white ginseng? Specific results should be enumerated.

Response) Thank you for pointing this out. In this study, the different in volatile compounds of ginsengs were not clear depending on cultivation ages. As you can see in Fig. S1, it is hard to distinguish the groups between the Panax species and cultivars. We tried to figure out whether the age of cultivation affect the volatile compounds. The results showed that the cultivation ages were not clustered in each ginseng group, which means the volatile composition of the ginseng roots were not affected by different cultivation ages (Fig. S6). We added this result in 3.2 section with Fig. S6.
